# Chemically-Boosted Corneal Cross-Linking for the Treatment of Keratoconus through a Riboflavin 0.25% Optimized Solution with High Superoxide Anion Release

**DOI:** 10.3390/jcm10061324

**Published:** 2021-03-23

**Authors:** Cosimo Mazzotta, Marco Ferrise, Guido Gabriele, Paolo Gennaro, Alessandro Meduri

**Affiliations:** 1Departmental Ophthalmology Unit and USL Toscana Sud-Est, 53100 Siena, Italy; cgmazzotta@libero.it; 2Post Graduate Ophthalmology School, Siena University, 53100 Siena, Italy; 3Siena Crosslinking Center, Via Sandro Pertini 7, 53100 Siena, Italy; 4Studio Oculistico Ferrise, 88046 Lamezia Terme, Italy; 5Department of Oral and Maxillofacial Surgery, “Le Scotte” Hospital, Viale M. Bracci, 53100 Siena, Italy; guid.gabriele@student.unisi.it (G.G.); paolo.gennaro@unisi.it (P.G.); 6Unit of Ophthalmology, Department of Biomedical Sciences, Dentistry, Morphological and Functional Imaging, University of Messina, 98100 Messina, Italy; alessandro.meduri@polime.it

**Keywords:** accelerated cross-linking, cross-linking, epithelium-off, keratoconus, riboflavin, SafeCross

## Abstract

The purpose of this study was to evaluate the effectiveness and safety of a novel buffered riboflavin solution approved for corneal cross-linking (CXL) in progressive keratoconus and secondary corneal ectasia. Following the in vivo preclinical study performed on New Zealand rabbits comparing the novel 0.25% riboflavin solution (Safecross^®^) containing 1% hydroxypropyl methylcellulose (HPMC) with a 0.1% riboflavin solution containing 0.10% EDTA, accelerated epithelium-off CXL was performed on 10 patients (10 eyes treated, with the contralateral eye used as control) through UV-A at a power setting of 9 mW/cm^2^ with a total dose of 5.4 J/cm^2^. Re-epithelialization was evaluated in the postoperative 7 days by fluorescein dye test at biomicroscopy; endothelial cell count and morphology (ECD) were analyzed by specular microscopy at the 1st and 6th month of follow-up and demarcation line depth (DLD) measured by anterior segment optical coherence tomography (AS-OCT) one month after the treatment. We observed complete re-epithelization in all eyes between 72 and 96 h after surgery (88 h on average). ECD and morphology remained unchanged in all eyes. DLD was detected at a mean depth of 362 ± 50 µm, 20% over solutions with equivalent dosage. SafeCross^®^ riboflavin solution chemically-boosted corneal cross-linking seems to optimize CXL oxidative reaction by higher superoxide anion release, improving DLD by a factor of 20%, without adverse events for corneal endothelium.

## 1. Introduction

Keratoconus is an ectatic disease of the cornea, characterized by biochemical and biomechanical instability of stromal collagen, leading to variation in posterior and anterior corneal curvatures and contemporary reduction of corneal thickness, with progressive deterioration of visual acuity due to increasing high order aberrations and irregular astigmatism [1,2]. The advent of corneal collagen cross-linking in the ophthalmological panorama of the last decade [3,4] and of cross-linking combined therapy with customized excimer laser ablation (also known as central cornea remodeling—CCR), poly(methyl methacrylate) (PMMA) intra-stromal ring segments (INTACS; Ferrara Rings) [5,6,7], and phakic intraocular lenses (pIOLs) [8] has transformed the conventional therapy of keratoconus, based, at best, on a lifetime of rigid gas permeable (RGP) contact lenses wearing or, at worst, on corneal transplant, improving its conservative management, thus reducing the necessity of corneal lamellar and penetrating graft between 30 and 50% according to the most recent literature [9].

Riboflavin UV-A corneal CXL represents a conservative and minimally invasive therapeutic approach to increase the biomechanical and biochemical stability of the corneal stromal tissue, thus contrasting primary and secondary ectasia progression. Different methods of enhancing the CXL of corneal collagen have been comparatively evaluated in the literature, and the association between riboflavin and UV-A at 370 nanometers (nm) wavelength has been found to be the most effective by significantly increasing the corneal stiffening and reducing toxicity in vivo, by preserving corneal transparency and sparing the endothelium, the crystalline lens, and the macula structures [10]. Spoerl et al. [11] were the first to investigate the use of riboflavin and UV-A irradiation to achieve corneal CXL. Vitamin B_2_ (riboflavin) is a hydrophilic molecule that plays two key roles in the CXL process. The first fundamental role is the photo-sensitizing effect, promoting the production of reactive oxygen species (ROS) or free radicals, such as singlet oxygen or superoxide anion, thus mediating the CXL reaction between adjacent collagen fibers and proteoglycans constituents of the stromal extracellular matrix [3,10,11,12].

The release of ROS from the activated riboflavin molecules induces the typical photo-dynamic type II reaction (oxidative deamination of corneal collagen), leading to the formation of molecular bridges (cross-links) between and within collagen fibrils and between corneal collagen fibrils and extracellular matrix proteoglycans. The second fundamental role of riboflavin in the CXL therapy is the UV-A photo-absorption, thus allowing the concentration of ultraviolet light A energy impact delivered by the UV-A source in the anterior half of the corneal stroma. The so-called “Riboflavin shielding effect” ensures the safety of corneal endothelium, lens, and retina from UV damage [13].

The biomechanical effect of CXL is oxygen-dependent; this dependency is of particular importance in high-fluence or accelerated CXL (ACXL) [14,15,16,17,18,19,20,21], and trans-epithelial or epithelium-on CXL and currently major protocol modifications and nomograms are being evaluated to maintain the efficiency of the technique compared to the conventional CXL technique [22,23,24,25].

One notable advancement in the field of riboflavin solution for CXL has been the use of dextran-free riboflavin since the hyper oncotic effect of dextran has been strongly associated with an intraoperative reduction of corneal thickness caused by stromal dehydration and inter-lamellar compaction, increasing the risk of stromal wound-related complications and exposing the endothelium to radiation damage [26].

The introduction of alternative riboflavin diffusion media, such as the hydroxypropyl methylcellulose (HPMC) instead of dextran, has been proven to avoid the corneal thinning caused by corneal dehydration, reported in the literature, with dextran solutions without any ophthalmic safety and tolerability issues [27], as further confirmed by intraoperative optical coherence tomography studies [28].

The concentration and composition of the riboflavin solution play a significant role in the clinical outcome of the procedure and the resulting demarcation line depth (DLD). In fact, numerous factors, such as exposure time, mode of exposure (pulsed or continuous light illumination), UV-A irradiance, riboflavin solution formula, diffusion properties, drops administration, illumination beams, beam focus, and environmental conditions, might contribute to the variability of clinical outcomes [23]. Riboflavin concentration may range from 0.1% to 0.25%; increasing concentration from just 0.1 to 0.15% induces a variation of −22 µm in the depth of the demarcation line. Conversely, hypotonic riboflavin solutions increase the demarcation line depth to +22 µm [23].

It is known that an adequate concentration of riboflavin in corneal stroma is imperative to obtain a biomechanical effect of corneal CXL [29,30]. Some authors demonstrated that in vitro stromal concentration of riboflavin increased with exposure only if the epithelium was removed, which was ascribed to the impermeability of the corneal epithelium for substances with a molecular weight greater than 100 Dalton (Da) (riboflavin has a molecular weight of 338 Da) [31].

Since hydrophilic riboflavin cannot penetrate the intact hydrophobic corneal epithelium due to its chemical properties, in current standard CXL epithelium-off (Epi-off) techniques, the central corneal epithelium is mechanically debrided to allow sufficient riboflavin stromal imbibition with potential complications [32,33,34].

To overcome this issue, different strategies, such as different riboflavin formulations that would enhance its penetration through the intact epithelium, as well as prolongation of imbibition time, were developed. However, imbibition-time prolongation exposes corneas to excessive dehydration and thinning. Furthermore, deeper penetration of riboflavin can lead to increased corneal endothelium susceptibility to UV-A toxicity [35].

To reduce imbibition time, the iontophoresis-assisted riboflavin administration (I-CXL) has been introduced, taking advantage of riboflavin’s small molecular weight, water solubility, and negative charge—all useful properties in the iontophoresis process [36]. Moreover, a specific hypo-osmolar charged riboflavin, dextran-free, solution that uses EDTA 0.1% and trometamol 0.05% as enhancers has been introduced to optimize riboflavin stromal penetration by iontophoresis while reducing its side effects [37].

In 2015, an interesting study compared stromal riboflavin concentration after three corneal cross-linking (CXL) imbibition procedures: standard (Epi-off), transepithelial corneal (Epi-on), and iontophoresis-assisted technique (Ionto), using 0.1% hypotonic riboflavin phosphate in human corneas excised during deep anterior lamellar keratoplasty. The authors found that both transepithelial CXL techniques in combination with hypotonic-enhanced riboflavin formulation (RICROLIN+) were inferior to the standard CXL technique with riboflavin passive diffusion after epithelial removal [30].

In 2020, iVis Technologies S.r.l. (Taranto, Italy) introduced the SafeCross^®^ riboflavin solution, a European Community (CE) marked medical device specifically designed for corneal CXL in thin corneas (400 µm and under) and simultaneously combined technique of CXL and trans-epithelial excimer laser Central Corneal aberrometric Remodeling (CCR) [38,39,40,41]. This article reports the first clinical use of this novel riboflavin solution (SafeCross^®^) for riboflavin UV-A corneal CXL.

## 2. Materials and Methods

### 2.1. SafeCross^®^ Physio-Chemical Characteristics

SafeCross solution (iVis Technologies, Taranto, Italy) received CE approval on 20 February 2020 as a sterile ophthalmic solution for corneal cross-linking. The medical device was approved and indicated for Epi-off CXL procedures in primary (keratoconus, pellucid marginal degeneration) and secondary progressive ectatic corneal disorders (iatrogenic ectasia). As displayed in the medical device summary box in Table 1, the solution contains sodium riboflavin phosphate, corresponding to an equivalent of 0.25% (2.5 mg/mL) riboflavin, 1% (10 mg/mL) hydroxypropyl methylcellulose (HPMC), in a buffered solution containing sodium chloride, sodium tetra-borate 10H_2_O, and boric acid. The medical device was designed and indicated for faster penetration and diffusion into the corneal stroma during CXL procedures for progressive keratoconus and secondary ectasias. It is sterile, highly purified, transparent, non-irritating, yellow ocular viscoelastic isotonic solution with a high concentration of riboflavin and an optimized osmolarity (260–280 mOsm/kg), thus avoiding corneal intraoperative dehydration, capable of producing an amount of superoxide anion higher than 1.50 µmol/mL when irradiated with 5.4 J/cm^2^ fluence using a UV-A 365 nm UV-A power source.

The photochemical reduction of the riboflavin, which produces the superoxide anion, was monitored by analyzing the nitro-blue tetrazolium (NBT) reduction, which is converted in blue formazan (BF). The BF was spectrophotometrically determined at 580 nm using a modified method developed by Beauchamp and Fridovich [42]. The riboflavin solutions, in the presence of NBT, were exposed to UV-light at 365 nm with irradiation of 3 mW/cm^2^ for 30 min at a distance of 5 cm. The laboratory test was conducted in a dark environment to avoid light interference being riboflavin light-sensible. To show the higher amount of superoxide anion, the Safecross^®^ formula was compared to an aqueous solution with standard 0.10% riboflavin and 1% HPMC. The UV-A light irradiation induces excitation of the riboflavin to the triplet state, causing, in high O_2_ presence, the generation of superoxide anion. The superoxide anions react with carbonyl groups of the collagen chain, increasing the protein cross-linking. Thus, up to a certain constraint, higher riboflavin concentrations and higher dissolved oxygen concentrations of the solution synergically increase the yield of the reaction. Moreover, the addition of high molecular weight HPMC in the riboflavin solution and the use of low salt concentrations (oxygen in water decreases as a function of salt concentration) facilitate the absorption and transport of oxygen in the moisture [43,44,45]. At the low moisture content of HPMC (1%), most of the water molecules are present as HPMC bound water, which act as a plasticizer in the HPMC film, promoting the sorption of the oxygen in the film [46].

### 2.2. Preclinical In Vivo Study in Rabbits

The safety and performance of the SafeCross^®^ solution were firstly analyzed in a preclinical setting on rabbit eyes and then in a clinical setting in humans at the Siena Crosslinking Center (Siena, Italy) on the eyes of patients affected by progressive keratoconus scheduled for the Epi-off riboflavin-UV-A CXL. The end-points of the in vivo preclinical comparative study on rabbits were the evaluation of corneal re-epithelialization time and status, wound-related complications (haze) development in a 7-day follow-up. The in vivo comparative preclinical study comprised of 20 eyes of 10 New Zealand rabbits weighing 3.0 kg ± 0.6 kg and treated with epithelium-off CXL using 0.25% SafeCross^®^ and 1% HPMC solution on one eye and a 0.1% riboflavin + 0.10% EDTA solution on the contralateral eye. The corneas were followed-up for 7 days. Corneal debridement up to 8 mm diameter was executed with a refractive surgery blunt metal spatula. Stromal soaking was performed by instilling 1–2 drops of riboflavin solution per minute, for a total time of 10 min, until an adequate absorption of the riboflavin solution in the stromal tissue was obtained and verified by a portable slit lamp. The UV-A treatment was carried out by means of a CCL-VARIO corneal cross-linking UV-A emitter (Peschke Meditrade GmbH, Huenenberg, Switzerland) at a wavelength of 365 nm and a 9 mm spot diameter, with a 9 mW/cm^2^ UV-A power for 10 min of UV-light exposure. The total energy dose delivered was 5.4 J/cm^2^ at a 5 cm distance from the corneal plane. During the UV-A irradiation time, the cornea was furtherly soaked by applying 1–2 drops of riboflavin solution every 2.5 min for 3 times (at minute 2.5, at minute 5, and at minute 7.5).

At the end of the treatment, the cornea was washed with the sterile saline balanced solution (BSS), medicated with an antibiotic-steroid combination (tobramycin-dexamethasone ointment). The cornea was examined on each day from day 1 till day 7 after CXL. To evaluate the epithelial regrowth, the eyes were stained with topical fluorescein on each day following CXL and analyzed.

The phases of the preclinical study on rabbits are summarized in the following [Fig jcm-10-01324-ch001]:

Corneas remained transparent throughout the healing process in the SafeCross^®^-treated eyes (Figure 1a), while eyes treated with riboflavin + 0.10% EDTA showed a grade 2 to 3 haze in 55% of the cases (Figure 1b).

Seven days after surgery, corneas treated with SafeCross^®^ solution were not stainable with a fluorescence dye, showing a complete re-epithelization (Figure 2a), while positive fluorescence staining persisted in 40% of corneas treated with riboflavin + 0.10% EDTA (Figure 2b).

### 2.3. Clinical Study

The prospective interventional SafeCross^®^ clinical study in humans was approved by Siena Crosslinking Center Institutional Review Board following the tenets of the Helsinki declaration and comprised of 20 eyes of 10 patients with 6-month follow-up. Ten eyes (the worst eye affected by progressive keratoconus) were treated with standard epithelium-off CXL, and 10 (the fellow untreated eyes) were used as a control for endothelial morphology evaluation (pleomorphism and polymegathism). Specific informed consent was obtained from all subjects involved in the study.

Inclusion criteria: Patients with documented progressive keratoconus during the last 6 months before the treatment. Progression was established by subjective visual decay of at least one line associated with an increase in subjective astigmatism and/or myopia by 1 Diopter (D); increase in tomographic average Sim K by at least 1 D or more; decrease in minimum corneal thickness by at least 10 µm between two consecutive examinations at 6 months follow-up interval; minimum corneal thickness of no less than 400 µm at the thinnest point measured by blue laser scanning micro-slit corneal tomography (Precisio^®^, iVis Technology, Taranto, Italy); Amsler-Krumeich’s keratoconus classification stages II. Age range 21–35 years old.

Exclusion criteria: Severe dry eye; the history of herpes virus keratitis; concurrent corneal infections; previous ocular surgery; corneal opacities or scars; pregnancy.

Surgical technique: The accelerated CXL (ACXL) protocol using 9 mW UV-A power with 5.4 J/cm^2^ fluence was used for 10 min of UV-A irradiation in a 9 mm diameter of corneal surface after epithelial removal by a blunt metal spatula. Corneal soaking was performed with 0.25% riboflavin + 1% HPMC SafeCross^®^ solution for 10 min, administering 2 drops each minute. The dropping interval during the irradiation phase was 2 drops every 2.5 min (at time zero, 2.5 min, 5 min, 7.5 min). To reduce the risk for UV-exposure of the internal eye structures, miosis was induced preoperatively by applying one drop of pilocarpine 2%, 20 min before the treatment. The surgical technique is displayed in Table 2.

Standardized examinations were performed at baseline, at 1, 3, and 6 months after the procedure. At the end of UV-A illumination, the cornea was rinsed for 15 s with sterile balanced saline solution (BSS), medicated with tobramycin-dexamethasone eye-drops, cyclopentolate eye-drops, and dressed with a therapeutic soft contact lens bandage (Shalcon, Lab, Rome, Italy) to protect the corneal surface and help re-epithelization. Patients were re-examined at slit lamp on day 4, evaluating the re-epithelialization by a fluorescein dye test, bandage contact lens removal, and full bio-microscopic evaluation. Endothelial cell count was performed before and 30 days after the treatment with specular microscopy (Perseus, CSO, Florence, Italy) together with anterior segment corneal spectral domain Optical Coherence Tomography (OCT) (I-Vue, OptoVue, Freemont, Irvine, CA, USA) to detect the depth of the demarcation line. The degree of subjective discomfort was evaluated postoperatively on day 4 using a visual analog scale (VAS).

## 3. Results

The solution had a faster diffusion into the corneal stroma, decreasing the soaking time to less than 10 min (8 min were enough in all cases to achieve optimal imbibition). The possibility to shorten the riboflavin soaking time was particularly favorable, being the entire CXL procedure less time-consuming. SafeCross^®^ ensured an optimal viscosity and stability of the pre-corneal biofilm during the treatment for at least 3 min; see Figure 3a,b.

All eyes were re-epithelialized at day 4 when the therapeutic contact lens was removed, see Figure 4a,b. The average value reported on the VAS pain scale was 2 (range 1–3: mild discomforting).

No eye presented signs of endothelial damage at the postoperative control day 30, also compared with the fellow untreated eye; cell count and morphology remained well within normal parameters in the treated eyes. Preoperative mean endothelial cell density in treated eyes was 2342 cells/mm^2^ (range, 2042–2692 cells/mm^2^) and 2456 cells/mm^2^ in the control eyes (range 2172–2912 cells/mm^2^) without statistically significant differences and in the absence of morphological alterations. Postoperative endothelial cell count at 1 month was 2252 cells/mm^2^ on average (range 1987–2497 cells/mm^2^) and 2316 cells/mm^2^ on average (range 2089–2599 cells/mm^2^) at 1 and 6th postoperative month, respectively, see Figure 5a,b and Table 3.

The percentage of 0.25% riboflavin, despite the higher release of superoxide anion provided by the SafeCross^®^ solution, assured the protection of the corneal endothelium from UV-A irradiation.

Morphologically, the chemically-boosted CXL, provided by this novel medical device, resulted in an increased depth of cross-linking demarcation line compared to standard riboflavin solutions up to 20%, thus enhancing the volume of the treatment and reasonably its efficiency through the higher release of superoxide anion granted from its chemical formula, boosting up the effect of CXL penetration. Indeed, the demarcation line depth (DLD), measured by spectral domain anterior-segment (AS) corneal OCT in the first postoperative month, was found at a mean depth of 355.8 with a range of 329 µm–380 µm (20% over a 0.1% riboflavin solution with 1% HPMC vehicle and with same irradiation parameters), see Figure 6 and Table 3.

Anterior segment optical coherence tomography at first postoperative month, showing the depth of the demarcation lines between a 0.1% riboflavin solution with 1% HPMC vehicle and after 9 mW/5.4 J/cm^2^ ACXL, Figure 7a, and 0.25% riboflavin plus 1% HPMC SafeCross solution, Figure 7b after the same ACXL protocol, shows a 20% of higher penetration in the SafeCross^®^ treatment, reasonably due to the higher riboflavin concentration, optimized osmolarity, and reduced salt concentration, releasing a higher amount of superoxide anion in the corneal stroma.

## 4. Discussion

Riboflavin has a fundamental role in CXL therapy as a photosensitizer for the production of reactive oxygen species (singlet oxygen), the shield of UV-irradiation for corneal endothelium, and lens and retina through UV-A radiation absorption at 370 nm waveband. The thickness of riboflavin biofilm is related to the viscosity of the solution. Recommended dropping of the solution in the soaking time of CXL procedure is every minute and, during the irradiation, every 2.5 min. Moreover, riboflavin doesn’t impair oxygen diffusion during irradiation time [47].

The main parameters of a riboflavin solution influencing the efficacy and safety of a cross-linking procedure are riboflavin concentration (%) = 0.10, 0.14, 0.15, 0.22, 0.25, where the efficacy is related, with a pseudo-parabolic function, to the percentage of riboflavin in the solution. There is, therefore, a consistent dose-response curve with higher concentrations of riboflavin achieving greater CXL efficacy, suggesting that the manipulation of riboflavin dosage as well as the UV-A protocol can be used to optimize CXL. Thus, in a known riboflavin concentration, the greater is the percentage of riboflavin, the higher the amount of cross-linking density and the protection of the corneal endothelium. However, the optimal stromal riboflavin dosage for CXL is yet to be determined [48].

Riboflavin levels higher than 0.3% in ophthalmic solution are generally not used because if the riboflavin concentration is furthermore increased, the corneal stiffening is reduced because of the predominance of the superficialization of the shielding effect. As a fact, the riboflavin at high concentrations blocks the irradiation light, which then cannot reach the deeper layers of the cornea [49]. The calculated impact on the depth of the demarcation line is −22 µm for +0.5% step of concentration [23].

Another important feature of riboflavin solutions for the CXL is the vehicles (additives), such as dextran or HPMC and their % = 0.1, 1.0, 1.1, 2.0 or more, having the function of facilitating the transportation of oxygen necessary for the CXL-induced oxidative reaction and the stability of the solution biofilm over the corneal surface [27].

As found in our preclinical study on rabbits, riboflavin solutions containing EDTA may increase the risk of postoperative corneal opacification if used in epithelium-off procedures. In fact, since EDTA-containing solutions were initially offered for epithelium-on procedures, where EDTA acted as an “enhancer” to loosen the epithelial barrier, they may be related, as reported in the literature [50], to excessive intraoperative stromal dehydration, which may lead to unforeseeable postoperative complications, such as corneal opacification (haze), corneal flattening, a notable postoperative corneal thinning, and an undesired hyperopic shift.

SafeCross^®^ is approved for epithelium-off CXL and ACXL, having an osmolarity between 255 and 280 mOsm and a 0.25% riboflavin concentration with 1% HPMC (10 mg/mL) as a vehicle to boost the superoxide anion production.

The percentage of riboflavin and the type and quantity of additives present in SafeCross^®^ allow the production of a quantity of superoxide anion greater than 1.50 µmol/mL 5.4 J/cm^2^, i.e., 30% higher than other riboflavin solutions available on the market [42]. To the best of our knowledge, this is the unique solution available in the market and approved for Epi-off CXL with 0.25% concentration and an osmolarity between 260 and280 mOsm/kg that has been optimized to minimize the risk of alteration of the stromal structure, avoiding dehydration during the entire CXL procedure. According to preclinical studies and clinical applications, this novel solution used for accelerated CXL has guaranteed the safety of the corneal endothelium. The distribution of the solution biofilm on the corneal surface is stable and homogenous, thus avoiding the presence of break-up time spots and consequent endothelial hot-spots. Indeed, if break-up time spots are present, the damage thresholds may be exceeded locally, leading to a potential localized endothelial burn [13]. The 1% hydroxypropyl methylcellulose (HPMC) as riboflavin vehicle, instead of dextran, has long been established for ophthalmic safety and optimal tolerability [51,52,53,54]. Methylcellulose, indeed, has low surface tension and contact angle, which increases coating ability, having a lack of elasticity, which makes it more “visco-adherent” than visco-elastic [47,55]. It has been shown not to damage the endothelium [52] with many other advantages, such as its ability to be autoclaved, to have low cost, and allowing the possibility to be stored and shipped at an air-room temperature [55]. Moreover, as reported in the literature [28], the HPMC prevents the dangerous corneal thinning caused by corneal dehydration with dextran solution. Riboflavin and fluorescein have similar polarity and molecular weight (376 g/mol) with close diffusion coefficients (6 × 10^−7^ cm^2^/s at 35 °C) [12].

Since HPMC significantly increases the penetration of topical fluorescein as compared to other commonly used ophthalmic vehicles [56], it has been used as an optimal vehicle for riboflavin molecules, allowing their faster diffusion into the corneal stroma and decreasing the soaking time under 10 min (8 min on average ±1). The results on the riboflavin’s shortened soaking time are particularly favorable for patients and surgeons, as the entire CXL procedure is less time-consuming with relative advantages. As showed in this study, both in preclinical animal setting and in the pilot human clinical experience at the Siena Crosslinking Center in epithelium-off ACXL, the corneal re-epithelialization was rapid and completed after four days of soft contact lens bandage in all cases, and patients reported subjectively only a small degree of discomfort that was evaluated according to the VAS pain scale where the average value reported in our series was 2 (range 1–3: mild discomforting). Morphologically, the intrinsic capacity of the SafeCross^®^ in releasing a higher amount of superoxide anion into the corneal stroma during the Epi-off CXL resulted in an effectively increased depth of cross-linking (i.e., increased DLD), up to 20%, compared to standard 0.1% riboflavin plus 1% HPMC solutions, thus enhancing the treatment’s volume and boosting up the effect of CXL penetration. Indeed, as shown in Figure 7b, the average depth of the demarcation line at the AS-OCT at postoperative was found at a mean depth of 360 ± 30 µm (20% over a 0.1% riboflavin solution with 1% HPMC vehicle and with same irradiation parameters). According to its specific features, this aspect may represent a chemical enhancement of CXL through enhancing the depth of the demarcation line and thus the density of cross-linked tissue. The demarcation line depth doesn’t represent a simple morphological sign of CXL penetration, but it correlates, even not linearly, with CXL biochemical and biomechanical impact. The concept can be easily understood by the characteristic of the short-wave UV-mediated CXL, as reported in the biochemical studies by Weadock et al. [57]. From these investigations, it has been indeed clearly proved that collagen molecules possess many amino acid reactive residues, such as lysine and proline, constituting over the 80% of collagen amino acids that are recruited in the strong chemical (aldehydes-mediated) CXL, while only a limited amount (less than 20%) of free reactive residues of aromatic amino acids, such as tyrosine and phenylalanine, are involved in the short-wave UV-mediated CXL [58]. Thus, the cross-linking density can rise only up to an upper bound value, i.e., the saturation value. The cross-linking density in the superficial layers ends after reaching saturation and cannot be increased indefinitely, so its overall efficiency attains a homogeneous distribution not only into the stroma horizontally but also in a “depth-dependent curve” [59,60]. According to a recent review report on the significance of the demarcation line, the detection of a deeper demarcation line (i.e., higher CXL volume) is a positive factor in terms of safety for endothelium and overall biological efficiency [59].

## 5. Conclusions

SafeCross^®^ solution was intra and postoperatively well-tolerated, with no irritating, toxic, and allergic effects on the ocular surface, having a good surface distribution, optimal biofilm viscosity, and penetration into the corneal stroma.

According to its safety and efficacy in this preliminary study, the SafeCross^®^ solution may be proposed for a faster and homogeneous corneal soaking, avoiding the unpleasant and potentially endothelium-toxic effect of intra-operative corneal thinning occurring during the CXL treatment with standard riboflavin solutions containing high molecular weight dextran, offering a chemically-boosted CXL—unique in the panorama of the riboflavin solutions available on the market for Epi-off CXL.

This preliminary study presents some limitations, such as a low number of eyes and short postoperative follow-up time, so we hope that future studies with larger study cohorts will confirm these preliminary findings since every step forward we take in improving the safety and efficacy of our procedures is positive. After further studies, a possible new scenario of application of this solution could be in CXL plus treatments with sequential and simultaneous Customized Trans-Epithelial No-touch (cTen^®^) photo-ablation, corneal shape remodeling for the reduction of total high order aberrations through a specific customized interactive iVis Technologies proprietary software (CIPTA^®^) [61].

The use of this solution in thin corneas with a minimum corneal thickness between 350 and 400 µm (epithelium included) requires specific ACXL nomograms, such as the M nomogram developed by Mazzotta et al. for standardized pachymetry-guided ACXL [23].

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
