# Peer review of "Chemically-Boosted Corneal Cross-Linking for the Treatment of Keratoconus through a Riboflavin 0.25% Optimized Solution with High Superoxide Anion Release"

_jcm, 2021, doi:10.3390/jcm10061324_

Round 1

Reviewer 1 Report

Reviewer comments

This study evaluated the effect of a 0.25% riboflavin + HPMC % Safecross solution in corneal re-epithelialisation after collagen cross linking. An animal study was performed before the clinical study to evaluate its safety. The study reported positive results with quick absorption of the safecross solution through cornea in 8 mins; no signs of epithelial damage and complete re-epithealisation of the cornea post CXL.

Comments:

Intro: The introduction is very informative and provides the breadth of knowledge required to understand the paper.

Line 86: demarcation line Depth: remove full stop and use small “d’

Methods:

The methods section is difficult to follow and would suggest a flow chart of animal study and clinical study with timelines and number of eyes tested.

It will be good to move the results of the animal study to the results section of the paper.

Line 161: please correct typo to include use ‘of…’

The paragraph on “2.1. Safecross® Physio-Chemical CHARACTERISTICS “ is important but it would make it easier to read if it just provides a summary the characteristics or refence another paper. This is because the long section on characteristics takes the attention away from the methods of the study.

Figure 1a and 1b should be a part of the results.

Results:

I am curious as to why no statistics were performed as there was a contralateral eye. For e.g.: endothelial cell count.

It would help to include some distribution or Mean SD in the results, e.g.: soaking time.

It would be good to have some more info on the range of demarcation line depth (DLD) measured by anterior segment optical 24 coherence tomography (AS-OCT) on the eyes.

Line 246: remove “>”

Line 244 – 246: This sentence is like a conclusion. Please move this to the conclusion. While in the results please state what the data showed.

Line 245: “..having a good surface distribution, optimal biofilm viscosity and penetration into the corneal stroma. “- How was this evaluated?

Line 200- 201: how did the authors arrive at the result of 85%? Please move this section to results as it is the results of the animal study.

Table 1 and 2: please include footnotes to explain all acronyms used.

Figure 4a and 4b, show the re-epithealised cornea in white light and then under cobalt blue light. It would be more useful to see the before and after pics – with the epithelial defect on day 0 and then no defect on day 4.

It would be really useful to have a table for the 10 eyes + control and show the different parameters- absorption time, epithelial recovery time, cell count over time.

Discussion:

Line 307: not sure what the word ‘constrain’ is supposed to mean here.

Line 317- 322: The section on osmolarity is a sudden jump in the discussion as it has not been mentioned in the intro and is not a variable in the study per se.

Line 359- 360: The VAS again pops up suddenly and was not mentioned in the results.

It is a little difficult to navigate between the results and discussion. It will be helpful if the results and the discussion followed the findings.  

Author Response

On behalf of all authors, I would like to thank You all for the splendid comments and suggestions that will hopefully help us make a better work. We tried to reply to every comment as throughly as possible.

Intro: The introduction is very informative and provides the breadth of knowledge required to understand the paper.

 We are thankful to the reviewer for the positive comment.

Line 86: demarcation line Depth: remove full stop and use small “d’
Revised the typos as requested.

Methods:

The methods section is difficult to follow and would suggest a flow chart of animal study and clinical study with timelines and number of eyes tested.

We added a flow chart as requested for tha animal study and we added a table for the clinical study indicating the demarcation line depth and Endothelial cell count.

It will be good to move the results of the animal study to the results section of the paper.

Revised as requested.

Line 161: please correct typo to include use ‘of…’

Revised as requested.

The paragraph on “2.1. Safecross® Physio-Chemical CHARACTERISTICS “ is important but it would make it easier to read if it just provides a summary the characteristics or refence another paper. This is because the long section on characteristics takes the attention away from the methods of the study.

We understand that the paragraph 2.1 " Safecross® Physio-Chemical CHARACTERISTICS" is a long section and can be difficult to digest, but we think that it's important that the reader has all the available information at hand in a single paper, since physio-chemical characteristics can be a tough subject to elaborate on.

Results:

I am curious as to why no statistics were performed as there was a contralateral eye. For e.g.: endothelial cell count.

We specified in the review that the control group in the clinical study was used just for evaluation of pleomorfism and polymegathism.

It would help to include some distribution or Mean SD in the results, e.g.: soaking time.

Soaking time was the same in every eye treated, as specified in the manuscript.

It would be good to have some more info on the range of demarcation line depth (DLD) measured by anterior segment optical 24 coherence tomography (AS-OCT) on the eyes.

We agree with this point and we added a TABLE reporting all the demarcation line depths of the 10 eyes plus its range

Line 246: remove “>”

Revised as requested.

Line 244 – 246: This sentence is like a conclusion. Please move this to the conclusion. While in the results please state what the data showed.

We moved this sentence to the CONCLUSION section as requested.

Line 245: “..having a good surface distribution, optimal biofilm viscosity and penetration into the corneal stroma. “- How was this evaluated?

The formation of a stable corneal biofilm was evaluated by Break Up Time test at 10 minutes. Thickness of riboflavin is related to the viscosity and, as we know from literature (Wollensak G, Significance of the riboflavin film in corneal collagen crosslinking. JCRS 2010;36:114-20 ), HPMC solutions have a BUT of 10 minutes.

Line 200- 201: how did the authors arrive at the result of 85%? Please move this section to results as it is the results of the animal study.

We realized that the percentage of 85% was calculated uncorrectly and we corrected it to the real percentage of 40%. That means that of the 4 out of 10 eyes treated wit the riboflavin + 0.1% EDTA solution still presented a positive staining at the postoperative control.

Table 1 and 2: please include footnotes to explain all acronyms used.

The acronyms were revised as requested.

Figure 4a and 4b, show the re-epithealised cornea in white light and then under cobalt blue light. It would be more useful to see the before and after pics – with the epithelial defect on day 0 and then no defect on day 4.

We agree with this suggestion and we exchanged figure 4 a-b with a new figure 4 a-b-c-d illustrating the re-epithelization process during day 1-2-3-4 postop.

Discussion:

Line 307: not sure what the word ‘constrain’ is supposed to mean here.

We agree that the phrase was unclear so we modified it as requested.

Line 317- 322: The section on osmolarity is a sudden jump in the discussion as it has not been mentioned in the intro and is not a variable in the study per se.

We agree that this part can cause unnecessary confusion so we removed it.

Line 359- 360: The VAS again pops up suddenly and was not mentioned in the results.

We added the VAS in the results section as requested.

After reviewing our statistical analysis, we also corrected some data regarding endothelial cell count and demarcation line depth (DLD).

Reviewer 2 Report

The concept of this trial in assessing the safety and efficacy of Safecross as a high concentration and optimized Riboflavin solution for CXL was good. However, there are various concerns/questions:

  • Overall, too much detail is provided on biochemistry of Riboflavin and Safecross in the introduction and discussion, with minimum concentration on the trial and minimum comparisons between other studies assessing the efficacy and safety of high concentration Riboflavin in patients with progressive keratoconus. It is expected to find more info on other studies which tried higher strength Riboflavin.
  • The inclusion criteria in the human study should provide the minimum age of the patients. Also, it is vague in terms of the control group. Why was the fellow eye treated? Was there bilateral progression with one eye being worse than the other? Does that mean all 10 patients recruited in this study had bilateral progression? If not, how was the treatment of the fellow eye ethically justified?
  • There is confusion about the type of riboflavin in the control group in human study. In the rabbit study riboflavin 0.25% + 0.10% EDTA solution was used in the control group. In the human study, no information was provided about the type of the riboflavin in methodology. But then in the results there is a name of a riboflavin 0.1% solution with 1% HPMC.
  • The study was designed to compare the safety and efficacy of this new solution with currently available alternatives. However, there is no report of the corneal epithelialization and endothelial cell density in the control group.
  • LDL was used as an indicator of efficacy of Safecross. How about the mean and SD of LDL in the control group? How about a ttest between the groups with P value?
  • Cone flattening could be another indicator to assess the efficacy. How about the mean±SD of Kmax and ttest between the groups 6 months post-op?
  • Line 359: patient discomfort only mentioned in discussion and not in the methodology and results. Moreover, there is no comparison between the treatment and control groups.
  • There is a high number of self-citation: 13
  • Line 329: “therefore, 30% higher than other riboflavin solutions available on the market.” What is the reference?
  • Lines 395-401: it is concluded the Safecross could be used in combination with other corneal procedures. This conclusion cannot be inferred from this study.
  • Line 398: reference 66. There is no reference 66 in the bibliography.
  • What are the limitations of this study?
  • What are the suggestions for future studies?

Author Response

On behalf of all authors, I would like to thank You all for the splendid comments and suggestions that will hopefully help us make a better work. We tried to reply to every comment as throughly as possible.

The concept of this trial in assessing the safety and efficacy of Safecross as a high concentration and optimized Riboflavin solution for CXL was good.

We would like to thank the reviewer for the positive comment.

However, there are various concerns/questions:

Overall, too much detail is provided on biochemistry of Riboflavin and Safecross in the introduction and discussion, with minimum concentration on the trial and minimum comparisons between other studies assessing the efficacy and safety of high concentration Riboflavin in patients with progressive keratoconus. It is expected to find more info on other studies which tried higher strength Riboflavin.

Since this is the first study assessing the safety of SafeCross, we think it's reasonable to add as much as much detail as possible on this new riboflavin solution.

With this review we tried to expand on the trial part adding a Flow Chart for the Preclinical study in animals and a Table of all the results in the clinical stuudy in humans.

The inclusion criteria in the human study should provide the minimum age of the patients. Also, it is vague in terms of the control group. Why was the fellow eye treated? Was there bilateral progression with one eye being worse than the other? Does that mean all 10 patients recruited in this study had bilateral progression? If not, how was the treatment of the fellow eye ethically justified?

We added the range of age of patients in the manuscript.

In this preliminary study we didn't treat the fellow eye. We used the fellow eye just for comparison of endothelial layer characteristics such as polimegathism and pleomorphism. If after 6 months some of those patients showed significant progression of the keratoconus in the fellow untreated eye we eventually treated it (but it was not part of the study).

Since this passage of the study was unclear, we are thankful to the reviewer for making us notice, and with this review we tried to clarify this concept more according to this comment.

There is confusion about the type of riboflavin in the control group in human study. In the rabbit study riboflavin 0.25% + 0.10% EDTA solution was used in the control group. In the human study, no information was provided about the type of the riboflavin in methodology. But then in the results there is a name of a riboflavin 0.1% solution with 1% HPMC.

We didn't use any riboflavin in the fellow eye since it was not treated in this study, see response above.

The study was designed to compare the safety and efficacy of this new solution with currently available alternatives. However, there is no report of the corneal epithelialization and endothelial cell density in the control group.

We didn't treat the control group, please see response above.

LDL was used as an indicator of efficacy of Safecross. How about the mean and SD of LDL in the control group? How about a ttest between the groups with P value?

We added a table with postoperative month 1 control data on all eyes regarding both Demarcation line Depth and endothelial cell count.

No t-test between groups is possible since we didn't treat the contralateral eye in this study, please see respone above.

Cone flattening could be another indicator to assess the efficacy. How about the mean±SD of Kmax and ttest between the groups 6 months post-op?

The aim of this study was not to evaluate corneal curvature but to assess the demarcation line depth and changes in both morphology and density of the endotelial cells.

Line 359: patient discomfort only mentioned in discussion and not in the methodology and results. Moreover, there is no comparison between the treatment and control groups.

We agree and we added a line about VAS in both methods and results sections

There is a high number of self-citation: 13

We think that self-citations are an understandable consequence of our familiarity with our study group previous works about this subject, so we do not think they are unappropriate.

Line 329: “therefore, 30% higher than other riboflavin solutions available on the market.” What is the reference?

The average depth of the demarcation line using riboflavin 0.1% + HPMC 1% solutions can be found in literature, please see for example citation 23 ( Mazzotta C., Romani A., Burroni A. Pachymetry-based Accelerated Crosslinking: The “M Nomogram” for Standardized Treatment of All-thickness Progressive Ectatic Corneas. International Journal of Keratoconus and Ectatic Corneal Diseases, July-December 2018;7(2):137-144 )

Lines 395-401: it is concluded the Safecross could be used in combination with other corneal procedures. This conclusion cannot be inferred from this study.

We agree with this point and we corrected it as requested.

Line 398: reference 66. There is no reference 66 in the bibliography.

We corrected the reference as requested.

What are the limitations of this study?

We added the limitations in the conclusions as requested

What are the suggestions for future studies?

We added the suggestions in the conclusions as requested

Reviewer 3 Report

The sample size is too small and the follow-up is short, in order to draw scientific conclusions for this kind of clinical study. Only one eye per each patient should be analyzed for statistical analysis. The short-term follow-up is critical, since it is still unknown that chemically boosted CXL is effective for halting the progression of the disease.

Author Response

POINT 1: The sample size is too small and the follow-up is short, in order to draw scientific conclusions for this kind of clinical study. Only one eye per each patient should be analyzed for statistical analysis. The short-term follow-up is critical, since it is still unknown that chemically boosted CXL is effective for halting the progression of the disease.

RESPONSE 1: We agree with you that this is not a conclusive study. It is a work in progress. However it is the presentation to a preliminary record that paves the way to the future of chemically ehnanced corneal cross-linking, and we strongly believe that this new solution could open a new modality to increase the efficacy of cross-linkng metodology, together with other advancements (new protocols, customized fluences, oxygen, pulsed light).

Reviewer 4 Report

Vast research has been dedicated to different protocols for corneal crosslinking (CXL) in order to overcome its limitations and maximize its efficacy. Therefore, this is a hot topic and the Introduction section clearly highlights it. The manuscript is well written and properly organized, with an adequate English proficiency. The purpose of the study is well detailed, the methodology is appropriate, and results are well presented, both in text and in tables. While the Materials and Methods section is understandably extensive given the nature of the study and the inclusion of preclinical and clinical data, the Introduction section could be shortened.

Regarding UVA toxicity, apart from the endothelial susceptibility, there has also been some interesting research including microperimetry and OCT-angiography, that the authors can include in future research to take into consideration the possibility of macular toxicity.

Author Response

POINT 1: Regarding UVA toxicity, apart from the endothelial susceptibility, there has also been some interesting research including microperimetry and OCT-angiography, that the authors can include in future research to take into consideration the possibility of macular toxicity.

RESPONSE 1: Thanks for the very interesting suggestion, we will make sure to include this topic in future research.

Round 2

Reviewer 2 Report

Thank you for your modifications.

Based on your sample size, your comments on safety and efficacy of SafeCross (in result and conclusion section) should be provided in a cautious manner. Please amend your comments.

Line 357- please add reference

Line 358- please change funique to unique. There are more English spelling errors in the manuscript, which need correction.

Author Response

Thank you for your modifications.

On behalf of all authors, I would like to thank You all for the splendid comments and suggestions that will hopefully help us make a better work. We tried to reply to every comment as throughly as possible.

Based on your sample size, your comments on safety and efficacy of SafeCross (in result and conclusion section) should be provided in a cautious manner. Please amend your comments.

We agree that the comments in the results and conclusion section should be more cautious, so we revided this part as requested.

Line 357- please add reference

Revised as requested.

The photo-chemical reduction of the riboflavin which produces the superoxide anion, was monitored by analyzing the nitro-blue tetrazolium (NBT) reduction which is converted in blue formazan (BF). The BF was spectrophotometrically determined at 580nm using a modified method developed by Beauchamp and Fridovich.[43] The riboflavin solutions, in presence of NBT were exposed to UV-light at 365um with an irradiation of 3mW/cm2 for 30min at a distance of 5 cm. The laboratory test was conducted in a dark environment to avoid light interference being riboflavin light-sensible. To show the higher amount of superoxide anion, the Safecross® formula was compared to an aqueous solution with standard 0.10% Riboflavin and 1.0% HPMC. The UV-A light irradiation induces excitation of the riboflavin to the triplet state, causing, in high O2 presence, the generation of superoxide anion. The superoxide anions react with carbonyl groups of the collagen chain increasing the protein cross-linking. Thus, up a certain constraint, higher riboflavin concentrations and higher dissolved oxygen concentrations of the solution synergically increase the yield of the reaction. Moreover, the addition of high molecular weight HPMC in the riboflavin solution and the use of low salt concentrations (oxygen in water decreases as function of salt concentration) facilitates the absorption and transport of oxygen in the moisture.[44–46] At low moisture content of HPMC (1%), most of the water molecules are present as HPMC bound water, which act as plasticizer in the HPMC film, promoting the sorption of the oxygen in the film.[47

Line 358- please change funique to unique. There are more English spelling errors in the manuscript, which need correction.

Revised as requested.

Reviewer 3 Report

Again, the author does not correctly correspond to the first review. Please carefully read the comments and provide the point-by-point responses for them.

The sample size is too small and the follow-up is short, in order to draw scientific conclusions for this kind of clinical study. Only one eye per each patient should be analyzed for statistical analysis. The short-term follow-up is critical, since it is still unknown that chemically boosted CXL is effective for halting the progression of the disease.

Author Response

On behalf of all authors, I would like to thank You all for the splendid comments and suggestions that will hopefully help us make a better work. We tried to reply to every comment as throughly as possible.

The sample size is too small and the follow-up is short, in order to draw scientific conclusions for this kind of clinical study.

We agree with this point, so we added in the conclusions the following line in the conclusions: 

"This preliminary study presents some limitations, such as low number of eyes and short post-operative follow-up time, so we hope that future studies with larger study cohorts will confirm this preliminary findings, since every step forward we take in bettering the safety and efficacy of our procedures is positive."

In addition, we made some more modifications in the manuscript review
to try to better the overall quality of this work, such as:

-
We added a table with postoperative month 1 control data on all eyes regarding both Demarcation line Depth and endothelial cell count.

- We added a flow chart for a better readability of the animal study

- we exchanged figure 4 a-b with a new figure 4 a-b-c-d illustrating the re-epithelization process during day 1-2-3-4 postop.

Only one eye per each patient should be analyzed for statistical analysis.

In this preliminary study we didn't treat the fellow eye. We used the fellow eye just for comparison of endothelial layer characteristics such as polimegathism and pleomorphism. If after 6 months some of those patients showed significant progression of the keratoconus in the fellow untreated eye we eventually treated it (but it was not part of the study).

Since this passage of the study was unclear, we are thankful to the reviewer for pointing it out and with this review we tried to clarify this concept more according to this comment.

The short-term follow-up is critical, since it is still unknown that chemically boosted CXL is effective for halting the progression of the disease.

We agree that studies with a larger cohort of patients and longer follow-up will be necessary to confirm the long-term stability of the treatment, but the preliminary data available to us shows the safety of the treatment in the main morphological parameters that we wanted to evaluate in this study (e.g. no damage of endothelial cells, no stromal alterations or haze formation) as well as the clinical efficacy of the Accelerated 9mW Epi-Off Treatment with SafeCross and its morphological results, such as the increased average demarcation line depth (The average depth of the demarcation line using riboflavin 0.1% + HPMC 1% solutions can be found in literature, please see for example citation 23 Mazzotta C., Romani A., Burroni A. Pachymetry-based Accelerated Crosslinking: The “M Nomogram” for Standardized Treatment of All-thickness Progressive Ectatic Corneas. International Journal of Keratoconus and Ectatic Corneal Diseases, July-December 2018;7(2):137-144 )

Clinically, we found a flattening of the cone and a reduction on Kmax and Kmin aligned with other Epi-Off treatment protocols reported in literature and a stability of the ectasia, but since the aim of this paper is not the clinical anaysis but a morphological evaluation, we considered not to add the graphics regarding UCVA and BCVA – K readings – Sim K readings of corneal tomography since they need a longer follow-up, and concentrate on Demarcation Line Depth and Endothelial Cell Count.

We think that the possibility to use a (unique on the market) 0.25% riboflavin solution with increased release of super-oxide anion may help to compensate the theoretical lower distribution of oxygen in accelerated treatments (compared to the Standard 3 mW EpiOff treatment - Dresden Protocol), caused by the reduced time of treatment, with an increment of oxygen release due to the chemical properties of the SafeCross solution.